# Prolonged Antiretroviral Treatment Induces Adipose Tissue Remodelling Associated with Mild Inflammation in SIV-Infected Macaques

**DOI:** 10.3390/cells11193104

**Published:** 2022-10-02

**Authors:** Aude Mausoléo, Anaelle Olivo, Delphine Desjardins, Asier Sáez-Cirión, Aurélie Barrail-Tran, Véronique Avettand-Fenoel, Nicolas Noël, Claire Lagathu, Véronique Béréziat, Roger Le Grand, Olivier Lambotte, Christine Bourgeois

**Affiliations:** 1Université Paris-Saclay, Inserm (UMR 1184), CEA (IDMIT Department, IBFJ), Center for Immunology of Viral, Auto-Immune, Hematological and Bacterial Diseases (IMVA-HB), 92260 Fontenay-aux-Roses & 94276 Le Kremlin-Bicêtre, France; 2HIV Inflammation and Persistence, Institut Pasteur, Université Paris Cité, 75015 Paris, France; 3Université Paris-Saclay, AP-HP Hôpitaux Universitaires Paris Saclay, Service de Pharmacie, Bicêtre, Inserm (UMR 1184), CEA (IDMIT Department, IBFJ), Center for Immunology of Viral, Auto-Immune, Hematological and Bacterial Diseases (IMVA-HB), 94276 Le Kremlin Bicêtre, France; 4Université Paris Cité, INSERM, U1016, CNRS, UMR8104, Institut Cochin, APHP, Hôpital Cochin, Laboratoire de Virologie, 75014 Paris, France; 5Université Paris-Saclay, AP-HP Hôpitaux Universitaires Paris Saclay, Service de Médecine Interne, Bicêtre, (UMR 1184), CEA (IDMIT Department, IBFJ), Center for Immunology of Viral, Auto-Immune, Hematological and Bacterial Diseases (IMVA-HB), 94276 Le Kremlin Bicêtre, France; 6Inserm UMR_S938, Centre de Recherche Saint-Antoine, Institut Hospitalo-Universitaire de Cardio-Métabolisme et Nutrition (ICAN), Sorbonne Université, 75012 Paris, France

**Keywords:** adipose tissue, HIV, SIV, chronic inflammation, metabolic changes, immune alteration

## Abstract

During chronic SIV/HIV infection, adipose tissue (AT) is the target of both antiretroviral treatment (ART) and the virus. AT might subsequently contribute to the low-grade systemic inflammation observed in patients on ART. To evaluate the inflammatory profile of AT during chronic SIV/HIV infection, we assayed subcutaneous and visceral abdominal AT from non-infected (SIV−, control), ART-naïve SIV-infected (SIV+) and ART-controlled SIV-infected (SIV+ART+) cynomolgus macaques for the mRNA expression of genes coding for factors related to inflammation. Significant differences were observed only when comparing the SIV+ART+ group with the SIV+ and/or SIV− groups. ART-treated infection impacted the metabolic fraction (with elevated expression of PPARγ and CEBPα), the extracellular matrix (with elevated expression of COL1A2 and HIF-1α), and the inflammatory profile. Both pro- and anti-inflammatory signatures were detected in AT, with greater mRNA expression of anti-inflammatory markers (adiponectin and CD163) and markers associated with inflammation (TNF-α, Mx1, CCL5 and CX3CL1). There were no intergroup differences in other markers (IL-6 and MCP-1). In conclusion, we observed marked differences in the immune and metabolic profiles of AT in the context of an ART-treated, chronic SIV infection; these differences were related more to ART than to SIV infection *per se*.

## 1. Introduction

Although the life expectancy of people living with HIV (PLWH) has increased considerably since the introduction of antiretroviral therapy (ART), comorbidities unrelated to AIDS contribute significantly to HIV-associated morbidity and mortality [1]. It is increasingly acknowledged that systemic inflammation and persistent immune activation have important roles in the pathogenesis of these comorbidities in general and cardiovascular and metabolic complications in particular [2]. The chronic immune activation observed in PLWH appears to be multifactorial, with (i) residual viral replication in reservoirs, (ii) dysbiosis and microbial translocations related to loss of integrity of the digestive barrier, (iii) viral co-infections (hepatitis B and C viruses (HBV and HCV), Epstein-Barr virus, cytomegalovirus, etc..) and immune defects [3]. In view of the pro-inflammatory potential of adipose tissue (AT) and the recently characterized AT disturbances associated with HIV/simian immunodeficiency virus (SIV) infection, one can hypothesize that AT in PLWH is an additional cofactor in residual inflammation. Indeed, AT’s endocrine activity and pro-inflammatory properties have been well described in the context of obesity [4]. Various inflammatory processes operate in the AT of obese individuals: (i) elevated production of pro-inflammatory soluble factors, including hormones (such as leptin), cytokines (such as interleukin (IL)-6 and tumor necrosis factor alpha (TNF-α)), chemokines (such as the monocyte chemoattractant protein 1 (MCP-1, also known as CCL2) [4], and micro-RNAs [5,6]; (ii) perturbations in the partitioning of lipids between AT and other tissues, leading to the ectopic accumulation of lipids that contributes to inflammation [7]; and (iii) changes in the adipocyte and/or immune cell composition of AT, the extracellular matrix (ECM), and the stromal vascular fraction (e.g., the accumulation of macrophages with a pro-inflammatory phenotype [8], CD8 T cells [9], and CD4 T cells with a predominant Th1 and Th17 phenotype) [4]. The AT remodelling and fibrosis in obesity results in local inflammation that spreads systemically and leads to the development of obesity-related comorbidities.

The nature and mechanisms of the HIV-induced disturbances of AT have not been characterized in detail [10]. The link between HIV and AT has been studied intensively because of the metabolic changes observed in many PLWH (notably dyslipidemia, insulin resistance, and lipodystrophy) [11,12]. Although these changes were initially considered to be adverse drug reactions to first-generation antiretrovirals (ARVs), their occurrence in treatment-naïve PLWH suggests that HIV has an impact *per se* on AT [13]. Viral proteins modify adipocyte functions [14,15,16,17], which contributes to fat fibrosis, changes in adipogenesis and AT senescence; in turn, these changes might impair adipocyte function and contribute to insulin resistance. HIV’s in vivo contribution to changes in AT has been further evidenced by the persistence of the virus in resident CD4 T cells and (to a lesser extent) macrophages [18,19].

It is now known that HIV and/or ARVs affect the amount, distribution and composition of AT; these changes are associated with immune and metabolic disturbances. As observed in obesity, cellular changes in the AT occur during SIV [18] or HIV [20] infection. One of the main features is a change in the T cell profile, with a predominance of CD8 T cells and thus an inversion of the CD4/CD8 ratio (relative to non-infected individuals) [18]. In contrast, there are few published data on the inflammatory profile of AT in PLWH. In vitro studies have shown that co-culture with CD4 T cells containing proviral DNA increases preadipocytes’ IL-6 expression by almost 3-fold [19]. However, the available in vivo data are difficult to interpret because they concern the study of subcutaneous adipose tissue (SCAT) from obese PLWH with metabolic complications [21] or from lipodystrophic areas in patients on ART [22,23]; these SCAT samples featured the underexpression of adiponectin and leptin and the overexpression of IL-6 and TNF-α, relative to healthy donors [22]. Given the invasive nature of the biopsy, few studies have evaluated changes in visceral adipose tissue (VAT); however, it appears to be less affected than SCAT [24,25]. The AT inflammation associated with HIV infection thus remains to be characterized in detail.

To avoid bias due to interindividual differences in the metabolic background and infectious history of human samples (both of which impact AT biology [12,20]), we studied a non-human primate model of SIV infection treated effectively (or not) with ARVs; this model mimics the main features of HIV infection but enables infection variables, follow-up times, the ART modality, treatment adherence, and bodyweight to be closely controlled. We performed transcriptomic analysis of SCAT and VAT samples from uninfected (SIV−) macaques, SIV-infected (SIV+) macaques, and ARV-treated, SIV-infected (SIV+ART+) macaques. The study’s primary objective was to determine the inflammatory signatures of SCAT and VAT from SIV-infected animals receiving (or not) ART.

We assessed AT inflammation by measuring the mRNA expression of soluble markers of inflammation associated with obesity (adiponectin, leptin, IL-6, TNF-α, MCP-1, CX3CL1, and CCL5) or viral infections (Mx1, CXCL10, and interferon-gamma (γ-IFN)). We also analyzed the mRNA expression of markers of changes in the cellular composition of AT: adipocyte-related genes (PPARγ and CEBP-α), hematopoietic markers (CD45, CD163, and γ-IFN), and ECM markers (COL1A2 and HIF-1α).

## 2. Materials and Methods

### 2.1. Animals, Infection, and AT Samples

Cynomolgus macaques (*Macaca fascicularis*) were imported from Mauritius and housed in the animal facility at the Commissariat à l’Energie Atomique et aux Energies Alternatives (CEA, Fontenay-aux-Roses, France); these non-human primates were housed and handled in accordance with French national regulations and were subject to inspection by the veterinary authorities (CEA authorization number: A 92-032-02). The CEA facility complies with the Standards for Human Care and Use of Laboratory of the Office for Laboratory Animal Welfare (OLAW, Bethesda, MD, USA) under OLAW assurance number #A5826-01. The use of non-human primates at CEA is also in line with the applicable European Directive (2010/63, recommendation Nu9). The macaques were studied under the supervision of the veterinarians in charge of the animal facility. For ethical reasons, animals were not sacrificed for the sole purpose of collecting AT. Samples from various studies (the SIVART, pVISCONTI and re-use programs) were analyzed in the present work, after the principal investigator of each program had given his/her consent.

As mentioned in the Introduction, we studied SCAT and VAT samples from three groups of adult male cynomolgus macaques: uninfected (SIV−) animals (*n* = 9), SIV-infected (SIV+) animals (*n* = 12), and SIV-infected ARV-treated (SIV+ART+) animals (*n* = 12). SIV+ and SIV+ART+ animals were administered the SIVmac251 strain intravenously at a dose of 1000 × the “50% animal infectious dose”, as previously described [18]. The animals assigned to the SIV+ART+ group were treated subcutaneously with a daily combination of three ARVs (emtricitabine (FTC, 40 mg/kg), tenofovir disoproxil fumarate (TDF, 5.1mg/kg) and dolutegravir (DTG, 2.5 mg/kg)) at week 4 (*n* = 7) or 24 (*n* = 4) after infection; this combination is currently used in the clinical context and includes 2 nucleoside reverse transcriptase inhibitors and 1 integrase strand transfer inhibitor (INSTI). Samples were collected during the chronic phase of the infection and then at euthanasia (i.e., 1-year post-infection for SIV+ animals and 2-years post-infection for SIV+ART+ animals) or from SIV− (control) animals. SCAT and VAT samples were collected from the abdominal area during necropsy. Non-adipose-associated tissues (including lymph nodes and blood vessels) were removed to prevent blood contamination. Samples were, cut into pieces weighing 50–200 mg, and stored at −80 °C under preservative-free conditions.

### 2.2. RNA Isolation and Quantitative Reverse Transcriptase-PCRs

Aliquots of 50 to 100 mg of AT were dissociated in Qiazol (Qiagen, Venlo, The Netherlands) by using a Tissue-lyser (Qiagen). mRNAs were purified with a GeneJET RNA Purification kit (Life Technologies, Carlsbad, CA, USA), according to the manufacturer’s instructions. The quality and concentration of the extracted RNA and the absence of contaminants (protein residues and organic solvents) were checked with a Nanodrop system. A reverse transcription step was used to produce complementary DNA (Enhanced Avian RT-PCR Kit, Sigma-Aldrich, St. Louis, MO, USA). Expression of the genes of interest was analyzed with real-time quantitative PCRs and specific primers (Life Technologies), the Taqman Universal PCR Master Mix (Life Technologies), and the BioRAd CFX96 Real Time System (50 amplification cycles, each corresponding to 15 s at 95 °C and 1 min at 60 °C). The limit of detection (LOD) was defined as a difference of 20 cycles between the level of expression of the gene of interest and the mean level of expression for the housekeeping genes (i.e., a 10^−8^-fold difference).

### 2.3. The Molecular Biology Panel

We used primers (TaqMan™ Gene Expression Assay, Life Technologies) designed for the following genes: *TNFA* (Mf02789784_g1), *IL6* (Mf02789322_m1), *MCP1* (Mf02787889_g1), *CX3CL1* (Rh03043051_m1), *CCL5* (Mf02788105_m1), *ADIPOQ* (Mf02788052_m1), *LEP* (Mf02788316_m1), *IFNG* (Mf02788577_m1), *CD163* (Mf02801556_m1), *MX1* (Mf00895608_m1), *CXCL10* (Mf02788358_g1), *CD45* (Hs00894716_m1), *PPARG* (Mf02787679_m1), *CEBPA* (Mf02914998_s1), *COL1A2* (Mf01028969_m1), *HIF1A* (Mf04356452_m1). The selected housekeeping genes encoded peptidylprolyl isomerase A (*PPIA*) (Mf04932064_gH), large ribosomal protein (*RPLPO*) (Mf04343677_g1) and 18S ribosomal RNA (rRNA) (Hs99999901_s1). Most of the primers were developed specifically for the cynomolgus macaque; the *CD45* and *CX3CL1* primers contained the human sequence but their cross-reactivity with cynomolgus macaque RNA was confirmed separately; the 18S rRNA primers were the same for all eukaryotic cells. The primers were labelled with fluorescein phosphoramidite, which enabled the amplification to be monitored after each cycle. Amplification curves (including a ΔCt measurement) were analyzed using CFX Maestro software (version 2.0, Bio-Rad, Hercules, CA, USA). All PCR reactions were performed individually. We used Normfinder software (Aarhus, Denmark, Department of Molecular Medicine at Aarhus University Hospital Skejby) to define the most appropriate combination of housekeeping genes for the present analysis [26]. The corrections for the two housekeeping genes were performed by dividing the expression of our genes of interest (1/(2^CT^) by the mean expression of the 2 housekeeping genes (1/(2^(Mean CT PPIA+18S)^) [27].

### 2.4. Statistical Analysis

The animals’ characteristics were quoted as the mean ± standard deviation (SD), and the mRNA expression levels were quoted as the median [interquartile range (IQR)]. Continuous variables of the three groups were compared using the non-parametric Kruskal–Wallis test associated with Dunn’s multiple comparison test. The comparison of SCAT and VAT value were performed using the Mann–Whitney test.

The threshold for statistical significance was set to *p* < 0.05. All statistical analyses were performed using GraphPad Prism 9 software (GraphPad Software, La Jolla, CA, USA).

## 3. Results

### 3.1. Characteristics of the Study Population

At necropsy, the SIV− (control) group had a mean ± SD age of 6.4 ± 1.2 years and a mean weight of 8.1 ± 1.2 kg. The SIV+ group had a mean age of 7.3 ± 2.4 years and a mean weight of 8.1 ± 3.6 kg. Relative to the SIV− group, the SIV+ART+ group was significantly older (mean: 8.1 ± 0.7 years, *p* = 0.0025) but did not differ with regard to weight (mean: 9.2 ± 1.0 kg). The SIV+ and SIV+ART+ groups did not differ significantly with regard to age and weight.

The animals were monitored for clinical and laboratory variables, as described elsewhere [28,29]. The infection peaked on day 12 and was followed by a plateau. The viral loads at euthanasia were 1 × 10^4^ [1.0 × 10^4^–5.9 × 10^4^] in the SIV+ group and <50 copies/mL in the SIV+ART+ group. Although the duration of infection differed in the SIV+ group (12 months) versus the SIV+ART+ group (24 months), both groups were observed during the plateau phase.

### 3.2. Housekeeping Gene Selection

AT is highly active metabolically and is also highly plastic in response to various signals [30]. The choice of the housekeeping gene is therefore crucial and depends on the pathophysiological context. We compared the mRNA expression of three housekeeping genes commonly used in studies of AT [31,32]: *18S*, *PPIA*, and *RPLPO* among SCAT and VAT of noninfected and infected animals (Figure 1).

In our analysis of housekeeping gene expression, each housekeeping gene’s mRNA expression level was normalized against the total amount of RNA. In the SIV− group, the expression level of *18S*RNA was significantly higher than those of *PPIA* and *RPLPO* in both SCAT and VAT. We did not detect a difference in expression between SCAT and VAT for any of the three genes. We next evaluated the stability of housekeeping gene expression in the three pathophysiological contexts of interest: SIV−, SIV+, and SIV+ART+ (Figure 1). We observed a trend towards (30-fold) lower expression of *18S* rRNA in the SIV+ART+ group (relative to the SIV− group) in both SCAT and VAT. Significant differences in the mRNA expression of *PPIA* were observed when comparing the SIV+ and SIV+ART+ groups (in both SCAT and VAT) and the SIV− and SIV+ART+ group (in VAT). In contrast to the results for *18S* rRNA, *PPIA* mRNA expression was higher in SIV+ART+ animals than in SIV− or SIV+ animals. The level of *RPLPO* mRNA expression differed least when comparing the three pathophysiological contexts but was always very low.

According to our NormFinder analysis of gene expression stability, none of the three genes had all the features required of a highly accurate housekeeping gene. Again, the gene with the most stable expression was *RPLPO* (stability value = 0.533), although the expression levels were very low. The best two-gene combination was *18S* + *PPIA* (stability value = 0.527) (Appendix A), and so we used it for subsequent experiments.

### 3.3. Changes in the Composition of AT following ART-Naive or ART-Treated SIV Infection

We investigated the composition of the AT in the SIV−, SIV+ and SIV+ART+ groups by focusing on specific markers of the adipocyte fraction (PPARγ and CEBPα), the hematopoietic fraction (CD45), and the ECM fibroblasts (COL1A2 and HIF-1α) (Figure 2).

We first measured the overall mRNA expression levels of the target genes in non-infected (SIV−) animals. The absolute expression level varied greatly from one target gene to another. In SIV− animals, *PPARG*, *CEBPA*, and *COL1A2* were expressed at much the same level as the housekeeping genes, whereas *CD45* was expressed at a much lower level (3000-fold lower) than *PPARG*; this low level of *CD45* expression (reported previously in human samples [33]) presumably reflects both the low number of CD45+ cells in AT and the relatively low level of *CD45* mRNA (due to extensive splicing and transcriptional modulation) [34]. To monitor the immune compartment more closely, we studied the mRNA expression of the surrogate markers *CD163* and *IFNG*, which are mainly expressed by anti-inflammatory macrophages and T lymphocytes/NK cells, respectively. The mRNA expression of *CD163* was significantly greater than that of *CD45*, whereas the mRNA expression of *IFNG* was low. Hence, the AT transcriptome content comprised high proportions of mRNA coding for genes expressed in adipocytes, fibroblasts and macrophages, whereas the transcripts associated with T cell activity were expressed at low but detectable levels. As expected, the transcriptomic signature was heavily weighted towards genes expressed in adipocytes.

We next evaluated the changes potentially associated with chronic SIV infection and ART. In the SCAT, *PPARG* expression was significantly greater in the SIV+ART+ group than in the SIV− and SIV+ groups (Figure 2a), whereas there was no difference between the SIV+ and SIV− groups. A similar profile was observed in the VAT: *PPARG* expression was greater in the SIV+ART+ group than in the SIV+ group, with no other intergroup differences (Figure 2b). The median *PPARG* expression level in the SIV+ART+ group was 43-fold higher than in the SIV+ group in SCAT and 8-fold higher in VAT. Since *PPARG* is predominantly but not exclusively expressed by adipocytes (the transcription factor is also expressed by Tregs [35]), we also studied the expression of *CEBPA*; this protein is also expressed late in the adipocyte maturation cycle, concomitantly with *PPARG* [36]. Although the intergroup differences in *CEBPA* expression were smaller than those observed for *PPARG*; similar results were obtained in SCAT for the SIV+ART+ group vs. the SIV− and SIV+ groups. The mRNA expression of *CD45* was heterogenous in both SCAT and VAT, and no significant differences were found between the three groups (Figure 2a,b). To circumvent such heterogeneity, we used additional, indirect markers of the T/NK cell and macrophage fractions. We observed a trend toward higher *IFNG* RNA levels in the SIV+ART+ group versus the SIV− group within the SCAT and in the SIV+ART+ group versus the SIV+ group within the VAT. *CD163* RNA levels were higher in the SIV+ART+ group when compared to both the SIV− and SIV+ groups in the SCAT. In the VAT, *CD163* expression was higher in the SIV+ART+ group than in the SIV+ group (Figure 2a,b). We also observed significantly greater *COL1A2* mRNA expression in the SIV+ART+ group than in the SIV− and SIV+ groups for both the SCAT (>2 log-fold), and the VAT at a lesser extent. In summary, ART for an SIV infection was associated with many metabolic, immune and ECM differences in the animals’ AT.

To further evaluate the potential impact of these changes on the balance between fibrosis and metabolic activity, we calculated the *PPARG/COL1A2* ratio. In the SIV+ART+, this ratio was lower in both SCAT and VAT, relative to the SIV− group, with even a reversal of the ratio in the SCAT (Figure 2c). It is noteworthy that the *PPARG/COL1A2* ratio tended to be lower in the SCAT than in the VAT among infected animals but not among SIV− animals, which suggest that the imbalance was greater in the SCAT. Lastly, we sought to determine whether expression of the profibrotic marker HIF1-α (expressed ubiquitously by immune, endothelial and metabolic cells) reflected hypoxia-related changes in the ECM [37]. *HIF1A* mRNA expression was greater in the SIV+ART+ group than the SIV+ group in the SCAT and VAT, differences with the SIV− group were only detected in the SCAT (Figure 2d).

In summary, we observed significant higher mRNA expression levels of *PPARG*, *CEBPA*, *CD163*, *COL1A2* and *HIF1A* in the SIV+ART+ group than in the SIV+ and SIV− groups in SCAT. A similar picture was observed in the VAT, although the intergroup differences were smaller.

### 3.4. The Inflammatory Profile of AT

We next studied the mRNA expression of genes coding for soluble markers of inflammation in AT, including cytokines specifically produced by adipocytes (such as adiponectin and leptin), more ubiquitous cytokines (IL-6 and TNF-α), and chemokines (MCP-1, CCL5, CX3CL1, and CXCL10).

We did not observe any difference between the SIV− and SIV+ groups with regard to *LEP* or *ADIPOQ* expression. In contrast, the median mRNA expression of *ADIPOQ* was higher in SIV+ART+ animals than in SIV− animals for both the SCAT (by a factor of 20) and the VAT (by a factor of 15). In the SCAT, the mRNA expression of *ADIPOQ* was also greater in the SIV+ART+ group than in the SIV+ group. The only significant intergroup difference in the mRNA expression of *LEP* was a 12-fold greater level in the SIV+ART+ group than in the SIV+ group, for the SCAT. (Figure 3a,b). To evaluate the adipocytes’ net inflammatory profile, we calculate the *ADIPOQ/LEP* ratio—a standard marker of metabolic activity in AT [38]. The ratio was higher in the SIV+ART+ group than in the SIV− group in the SCAT. (Figure 3c).

Overall, SIV infection did not appear to be associated with changes in *ADIPOQ/LEP* production in AT. However, the higher level of adiponectin mRNA found in SIV+ART+ than in SIV+ animals suggest that ART has a notable impact.

Regarding the pro-inflammatory cytokines, we did not observe any significant intergroup differences in the mRNA expression of *IL6* in SCAT or VAT (Figure 4a). The highly heterogenous *IL6* expression in SIV− animals prevented us from drawing any conclusions. In the SIV− group, the mRNA expression of *TNFA* was low or sometimes undetectable in SCAT and VAT. The expression levels were similar in the SIV+ and SIV− groups. However, the median mRNA expression of *TNFA* in the VAT was significantly greater in the SIV+ART+ group than in the two other groups (by a factor of 1000 vs. SIV− and 277 vs. SIV+) (Figure 4b).

In summary, we detected few differences in the AT’s production of these two cytokines following infection. Differences were observed for the SIV+ART+ group, with greater expression levels of *TNFA* (but not *IL6*) and the anti-inflammatory *ADIPOQ* than in the SIV− group.

Lastly, we studied the mRNA expression of genes coding for chemokines that contribute to the inflammatory processes by recruiting immune cells and prompting the latter to generate pro-inflammatory signals. With regard to the mRNA expression of standard markers of macrophage recruitment (*MCP1*, *CCL5*, and *CXCL10*), T cell recruitment (*CX3CL1* and *CXCL10*) and NK cell recruitment (*CX3CL1*), we did not observe any significant changes in SIV+ animals versus SIV− animals [39,40]. Although the significant difference in *MCP1* expression have been widely reported in the context of obesity [41], we did not find any intergroup difference for SCAT or VAT (Figure 5a,b). We found that the mRNA expression of *CCL5* was significantly (16-fold) greater in SIV+ART+ animals than in the SIV− group in SCAT and VAT (Figure 5a,b). With regard to *CX3CL1*, the median mRNA expression level was low in the SIV− group. However, much higher levels (by a factor of up to 633) were observed within the SCAT and the VAT in the SIV+ART+ group, relative to the SIV− and SIV+ groups. It is noteworthy that in the SIV− group and the SIV+ group, the mRNA expression of *CX3CL1* differed markedly from one animal to another. Similarly, mRNA expression of *CXCL10* was very heterogeneous within the groups. However, we observed greater expression in the SIV+ART+ group than in the SIV− group for VAT but not for SCAT.

In summary, we observed differences between the SIV+ART+ and SIV− groups in the mRNA expression of genes coding for various chemokines (except *MCP1*). In contrast, no differences between the SIV− and SIV+ groups were observed.

### 3.5. IFN-Stimulated Gene Expression in AT

AT is also characterized by its production of type I IFN, although the latter’s exact biological role in this tissue has yet to be fully characterized [42,43]. However, type I interferon is highly involved in antiviral responses and as such contributes to HIV/SIV-associated inflammation. We therefore investigated the expression of *MX1* (an interferon-stimulated gene (ISG)) in AT under different pathophysiological conditions. *MX1* mRNA expression in both the SCAT and VAT was significantly higher in the SIV+ART+ group than in the SIV− group (with fold increases of 157 in SCAT and 41 in VAT) and the SIV+ group (with fold increases of 77 in SCAT and 28 in VAT) (Figure 6).

In summary, we observed a more prominent ISG expression within the AT—especially in the context of ART following an SIV infection.

As summary, the changes in markers reflecting cellular composition, inflammatory profile and ISG expression of SCAT and VAT collected from SIV−, SIV+ and SIV+ART+ animals are presented in Figure 7. With regard to the impact of SIV infection *per se*, we did not observe significant differences in the expression of 16 selected genes encoding factors involved in inflammation. A comparison of the SIV+ART+ group with both the SIV− and SIV+ groups revealed significant or non-significant differences in most of the genes tested and suggested that AT alterations induced during ART treated chronic SIV are mostly related to ART.

## 4. Discussion

It has been well documented that the metabolic properties of AT are affected by both SIV/HIV and ART. We and others have also reported that AT is a reservoir for HIV and that its immune composition is altered during HIV/SIV infection [18,19]. However, data on the AT’s potential contribution to the inflammation associated with chronic HIV infection are scarce. A major obstacle to performing this type of study is the need to collect AT samples from homogenous groups of metabolically healthy individuals; the BMI, cardiovascular disease, type and duration of ART, age, and sex may drastically modify the AT’s pro- or anti-inflammatory profile and so must be taken into account. We, therefore, chose to investigate the inflammatory signature of AT in the setting of chronic SIV infection in cynomolgus macaques. The objective was to determine whether the AT contributes to systemic inflammation in this context, as has already been described for obesity. We initiated ART early in the course of infection (i.e., before the animals’ immune system was too severely impacted), as is currently recommended in humans; this approach modelled an ARV-induced decrease in viral load with mild immune system impairments and also allowed us to control most of the variables related to the infection (the viral strain, the dose inoculated, and the infection route) and the treatment (the SIV+ART+ animals received the ARVs that are currently used in first-line ART in the clinic).

In the first step in our study, we selected the housekeeping (reference) genes for use in subsequent experiments. Ideally, housekeeping genes should be expressed constitutively in all tissues (often because they have an essential role in the maintenance of cell function) and stably under different inflammatory and metabolic conditions; this is particularly important in the case of AT, in view of the latter’s cellular heterogeneity and plasticity. Many different reference genes have been used in literature reports. The use of *18S* is subject to debate because its expression in AT is not very stable [44]. Housekeeping genes like as *HPRT* or *GAPDH* are also inappropriate for studies of AT because their expression is modulated significantly by the metabolic context. *18S*, *PPIA* and *RPLPO* have been suggested for studies of AT studies [31]. Unfortunately, none of these three genes proved to be a robust reference in our study. *18S* was the most strongly expressed of the three but mRNA levels were severely impacted in some SIV+ART+ animals, suggesting inconsistencies between individuals. *RPLPO* was the most stable of the three genes tested but was expressed at a low level, which precluded its use in our experiments. *PPIA* exhibited an intermediate profile, showing intermediate level of expression, and changes depending on the infectious context that were maintained in a close range (approximately 2–4 fold differences). We therefore decided to use NormFinder software to evaluate the stability of the three candidate reference genes. We confirmed that *RPLPO* was the most stably expressed single gene but found that the combination of *PPIA* and *18S* gave the highest stability index. Hence, the average expression level of these two genes was taken as the reference value in subsequent experiments.

Regarding changes in the AT’s inflammatory profile during SIV infection and ART (Figure 7), the expression of *IL6* mRNA was similar in all three groups. *TNFA* mRNA was not detected in the AT of SIV− group but was present in the SIV+ART+ group; these results suggest that immune-mediated inflammation is present but at a low level in the AT in this setting. We also observed a higher expression of leptin, that was associated with an even higher expression of adiponectin during chronic treated infection; these two hormones exert both metabolic and immunoregulatory functions, their increase can therefore either directly reflect fat gain and lipid storage on ART [45] (in accordance with the increased *PPARG* mRNA expression) or the onset of an unexpected anti-inflammatory activity [46] Along with the pro-inflammatory cytokine signature and the anti-inflammatory hormone signature we observed only weak signs of AT infiltration by CD45+ (hematopoietic) cells. However, a non-significant increase in *IFNG* mRNA expression (a T and NK cell marker) suggested that T and NK cells were recruited and/or activated to some extent. The most notable changes were the significant increase in *CD163* mRNA expression in VAT and the non-significant increase in SCAT, which suggested that macrophages were recruited during ART and/or were further polarized to an anti-inflammatory profile. However, the expression of *MCP1* (usually described as being essential for macrophage recruitment in the context of obesity) was similar in all three groups of animals. Lastly, we observed greater mRNA expression of the chemokines *CCL5* and *CX3CL1*, which suggested that immune cell recruitment was engaged but did not greatly change the AT’s cellular composition.

In contrast, we observed several changes in the mRNA expression of metabolic markers (Figure 7). With regard to the AT’s composition, we found that the expression of the adipocyte differentiation marker *PPARG* was significantly greater in the SIV+ART+ group than in the SIV+ group; this might have reflected adipocyte hyperplasia and/or hypertrophy after a long period of infection and/or ART; this difference has been observed previously in simian models [47] and in humans [48] and is associated with a relative increase in adipocyte density. It should be noted that *PPARG* expression is not strictly specific for mature adipocytes: AT-resident Tregs also express *PPARG* [35] but there are few of these cells in primates [48]. The mRNA expression of *CEBPA* confirmed the stimulation of adipocyte functions. We also observed more prominent, indirect signs of fibrosis and hypoxia in the SIV+ART+ group than in the SIV+ group (i.e., greater expression levels of the fiber marker *COL1A2* and the hypoxia marker *HIF1A*, respectively). Furthermore, we found that the *PPARG/COL1A2* ratio (a marker of the AT’s metabolic status) was abnormally low in the SIV+ART+ group; this might correspond to dysfunctional AT growth, with an imbalance between adipocyte hyperplasia/hypertrophy and expansion of the surrounding collagen fibers. Researchers have observed changes in the AT’s cellular architecture in PLWH, with heterogeneous adipocyte size and an increase in fibrosis [45,49]. Moreover, the dysfunction was more prominent in the SCAT than in the VAT, suggesting that the former is more susceptible to the effects of an ART-treated SIV infection. It is well documented that despite the ARVs’ highly beneficial role in controlling HIV infection, these drugs have metabolic side effects [50]. Almost all ARVs are associated with changes in AT, such as weight gain, lipodystrophy, and changes in lipid trafficking and storage [22,51]. It has been suggested that ARVs promote hypertrophy and expansion of the AT, leading to not only weight gain but also mitochondrial dysfunction, greater oxidative stress, and fibrosis [10,52]. More recently, due to clinical concerns, the impact of integrase strand transfer inhibitors (INSTIs) on AT has been studied in particular [53]. SCAT and VAT from obese, INSTI-treated HIV-infected patients show higher levels of fibrosis than samples from obese, INSTI-naïve (non-infected) patients [10,54]. In addition to its harmful effect on adipocyte biology, fibrosis might also have an impact on immune cell migration; this would explain the limited recruitment even in the presence of a chemokine signal. Furthermore, HIV infection *per se* is involved in the modification and redistribution of AT in ARV-naive subjects, suggesting that ARVs and HIV have synergistic effects [49]. However, the weak inflammatory signature detected in SIV+ animals during the chronic phase of the infection argues in favor of a predominant impact of ART.

Lastly, we observed a change in the expression of *MX1* mRNA, a type I-IFN-stimulated gene, in the SIV+ART+ group. A prominent type I IFN signature has been reported previously in the context of obesity and appears to be a response to metabolic dysregulation [42,43]. Some researchers have suggested that the type I IFN pathway reveals the adipocytes’ “dormant” inflammatory potential [43]. However, this change in the adipocyte secretion profile necessarily involves a metabolic shift towards aerobic glycolysis. Thus, one can hypothesize that inhibition of aerobic glycolysis in the context of a treated HIV/SIV infection prevents the switch to a pro-inflammatory adipocyte phenotype, despite a type I IFN-rich environment. The mechanisms preventing the expression of the pro-inflammatory phenotype remain to be characterized but might result from (i) viral escape mechanisms that inhibit the initiation of an immune response by the adipocytes, (ii) the effect of ARVs (especially on the adipocytes’ mitochondria) and/or (iii) inhibition of aerobic metabolism by fibrosis-induced tissue hypoxia. Interestingly, *IFNG* was strongly expressed in VAT in the SIV+ART+ group, as were *TNFA* and *CXCL10*—suggesting that the duration of infection and/or ART significantly impacted the type II IFN pathway as well.

Overall, the changes were most prominent in the SIV+ART+ group. The metabolic changes were prominent, whereas only moderate immune/inflammatory changes were observed (Figure 7); this mild inflammatory profile differed markedly from that described in the context of obesity suggesting that the 2 processes involve different pathways of AT alterations. As mentioned earlier, we observed changes in the tissue-level inflammatory profile, which were sometimes even anti-inflammatory. The elevated mRNA expression of *ADIPOQ* and *CD163* and the higher ADIPOQ/LEP ratio are puzzling and prompt us to formulate several hypotheses. The creation of an anti-inflammatory environment might be due to ARV toxicity (preventing effective immune recruitment and/or activation) or specific regulation of the AT for limiting inflammation (which might be harmful for the tissue’s metabolic functions). Furthermore, a whole-tissue analysis might not be precise enough to reveal subtle changes in the immune profile. We might thus have underestimated the contribution of resident immune cells to the changes in AT seen during a chronic infection. Nevertheless, our whole-tissue results also revealed the limited pro-inflammatory influence of the immune compartment. Lastly, it is possible that post-transcriptional regulatory mechanisms modified the AT’s inflammatory signature.

Several of our findings indicate that the AT establishes compensatory mechanisms to prevent a harmful inflammatory response. On the one hand, the predominant cytokine at the tissue level was adiponectin, which is known to have an anti-inflammatory role [4]. On the other hand, the elevated expression of type I IFN has sometimes been described as a mechanism that protects against metabolic dysregulation in the context of obesity [42]. The AT’s ability to limit the inflammatory response to infection might explain why it can act as a reservoir for HIV and many other pathogens (e.g., *Mycobacterium tuberculosis*, *Leishmania*, cytomegalovirus, etc.) [13]. Additionally, the only pro-inflammatory cytokine overexpressed here was TNF-α, which reportedly has a more limited systemic action than IL-6 [55]—suggesting that the local AT inflammation was not directly associated with systemic effects. The heterogeneity in *IL6* expression (notably in the control group) also confirmed the highly plastic nature of AT inflammatory activity, and might partly explain the difficulty in identifying a pro-inflammatory signature in ART-naïve or ART-treated SIV infected macaques.

Lastly, the major metabolic dysregulation and the moderate inflammatory consequences are reminiscent of “metabolically healthy” obese patients, i.e., a subgroup of obese individuals who do not have a metabolic syndrome (i.e., no hypertension, hypertriglyceridemia, insulin resistance, or NASH). Indeed, up to 50% of obese patients might be “metabolically healthy” (depending the definition) and have less VAT, a smaller adipocyte size, less adipocyte fibrosis, less immune cell infiltration, and a higher plasma adiponectin concentration [56]. The role of adiponectin is probably crucial in limiting metabolic complications as it promotes adipocyte differentiation, insulin sensitivity, and lipid accumulation [57]. Thus, one can hypothesize that a similar phenomenon occurs during HIV/SIV infection, with metabolic changes but few immune and inflammatory consequences. However, it should be noted that “metabolically healthy” obese status is often temporary and that these individuals may develop complications during their follow-up, perhaps due to the exhaustion of some compensatory mechanisms.

Our study had several limitations. Firstly, our decision to study the effects of infection and treatment at the whole-tissue level prevented us from studying modifications of the stromal vascular fraction with sufficient precision. Secondly, the SIV+ animals included in the study did not show any signs of disease progression; the duration of the infection was rather short (1 year in the SIV+ group, 2 years in the SIV+ART+ group) which might explain (at least in part) the non-prominent inflammatory signature observed in AT. Thirdly, and despite upstream selection, the groups of macaques differed in some respects: the duration of infection and the age (the SIV+ART+ animals being older). However, the animals in the SIV+ART+ group were not exposed to uncontrolled viremia for as long as the SIV+ animals were. Moreover, the fact that the inflammatory signature was weak in SIV+ animals (in the chronic phase of the infection) and strong in the SIV+ART+ animals suggests that the ART had an impact. Fourthly, the study duration prevented us from drawing firm conclusions about the reportedly harmful long-term systemic effect of *MX1* overexpression [58]. Fifthly, our study was limited to white AT. In fact, recent research suggests a role for brown AT in the development of cardiovascular comorbidities, including in the context of HIV [10,59]. Sixthly, our analysis of the mRNA of a panel of genes obviously provided only a partial picture of changes in AT during infection. The analysis of other markers would have enabled us to monitor other immune cells, such as innate lymphoid cells. Lastly, the macaques studied here were selected on the basis of their weight and did not have apparent signs of obesity at baseline; this prevented us from investigating the potential impact of underlying obesity on the inflammatory signature of AT in the context of chronic HIV infection.

## 5. Conclusions

Whole-tissue analysis of gene expression in the AT from SIV−, SIV+ and SIV+ART+ animals revealed predominantly metabolic changes that were more prominent in the SIV+ART+ group than in the SIV− and SIV+ groups. ART after a chronic infection appears to be associated with (i) remodelling of the AT, changes in adipogenesis and the ECM, predominant fibrosis (associated with adipocyte hyperplasia/hypertrophy and hypoxia), a moderate impact on the T/NK fraction and a more sustained effect on the macrophagic component, (ii) a mild impact on the inflammatory profile, with a marked increase in the anti-inflammatory adipokine adiponectin, and (iii) a modification of the type I interferon signature (perhaps in response to metabolic perturbations) that was not accompanied by a pro-inflammatory cytokine signature.

All these abnormalities probably reflect both the metabolic toxicity of ARVs in AT and the adaptive mechanisms initiated by the AT to limit inflammation and thus preserve metabolic functions. On the basis of our results, AT appears unlikely to contribute to systemic inflammation via cytokine production. However, chronic HIV/SIV infection does impact the metabolic and immune functions of AT; these changes might alter the production of metabolites and/or micro-RNAs by AT which in turn might indirectly impact systemic inflammation.

## Figures and Tables

**Figure 1 cells-11-03104-f001:**
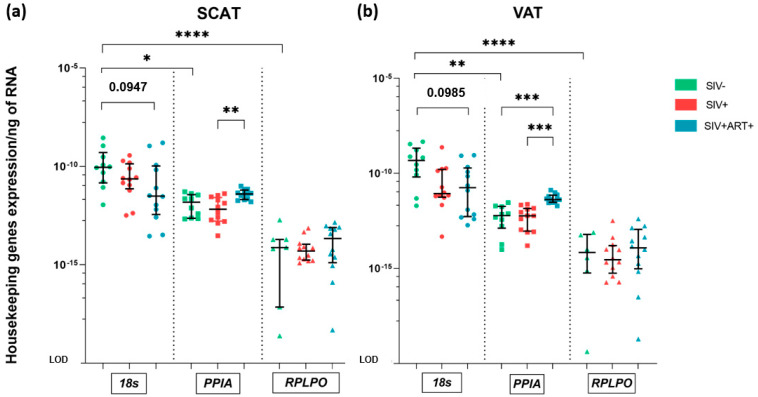
Housekeeping gene selection. Real-time quantitative PCR (RT-qPCR) analysis of the RNA expression of three housekeeping genes (*18S*, *PPIA*, and *RPLPO*). The level of gene expression was normalized against the total amount of RNA extracted from each sample. The limit of detection (LOD) corresponds to the difference after 50 cycles, i.e., a minimum detectable ratio of 10^−^^20^ after normalization against the total amount of RNA (**a**,**b**) RNA expression of *18S*, *PPIA* and *RPLPO* housekeeping genes within SCAT (**a**) and VAT (**b**) in the SIV− (green) (*18s* and *PPIA n* = 10, *RPLPO n* = 7), SIV+ (red) (*n* = 12) and SIV+ART+ (blue) (*n* = 12) groups. The results are quoted as the median [IQR]. Kruskal–Wallis tests associated with Dunn’s multiple comparison test or Mann–Whitney tests were used for unpaired and paired comparisons, respectively. Only statistically significant results are shown: * *p* < 0.05, ** *p* < 0.005, *** *p* < 0.001, **** *p* < 0.0001.

**Figure 2 cells-11-03104-f002:**
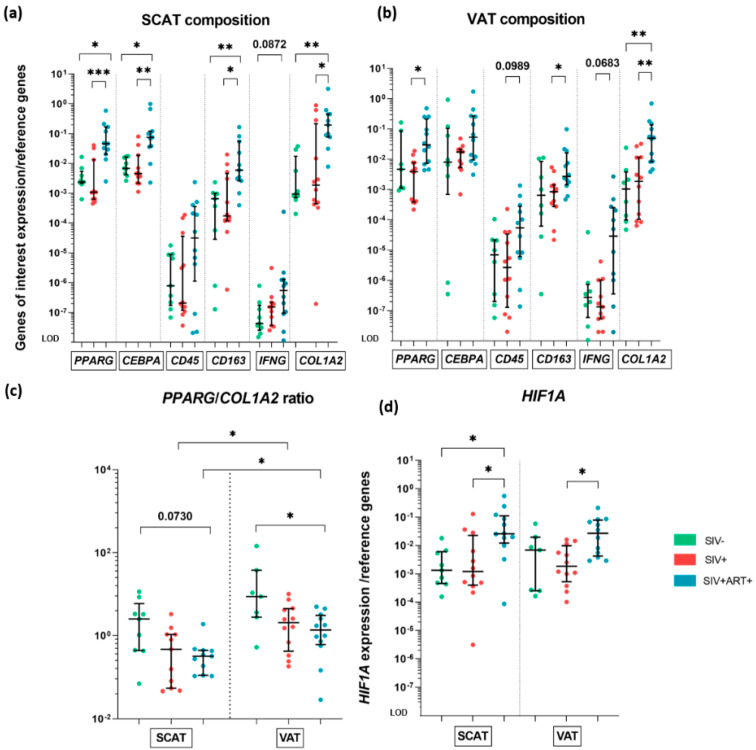
Gene expression analysis of AT components, at the whole-tissue level. RT-qPCR analysis of the mRNA expression of genes specifically transcribed in adipocytes (*PPARG* and *CEBP**A*), hematopoietic cells (*CD45*), immune cells (*CD163* and *IFNG*), and ECM (*COL1A2*) within SCAT and VAT and in the three groups of animals: SIV− (green), SIV+ (red), and SIV+ART+ (blue). Expression of the genes of interest was compared with that of the *18S* and *PPIA* housekeeping genes. The LOD was set to the difference vs. the mean level of housekeeping gene expression after 20 cycles, i.e., a 10 ^−^^8^-fold difference. (**a**,**b**) The composition of SCAT (**a**) and VAT (**b**) in the SIV−, SIV+ and SIV+ART+ groups. (**c**) The *PPARG/COL1A2* ratio in the SIV−, SIV+ and SIV+ART+ groups in SCAT (left) and VAT (right). (**d**) mRNA expression of *HIF1A* within SCAT (left) and VAT (right) in the SIV−, SIV+, SIV+ART+ groups. The results are quoted as the median [IQR]. Kruskal–Wallis tests associated with Dunn’s multiple comparison test or Mann–Whitney tests were used for unpaired and paired comparisons, respectively. Only significant results and trends are shown: * *p* < 0.05, ** *p* < 0.005, *** *p* < 0.001.

**Figure 3 cells-11-03104-f003:**
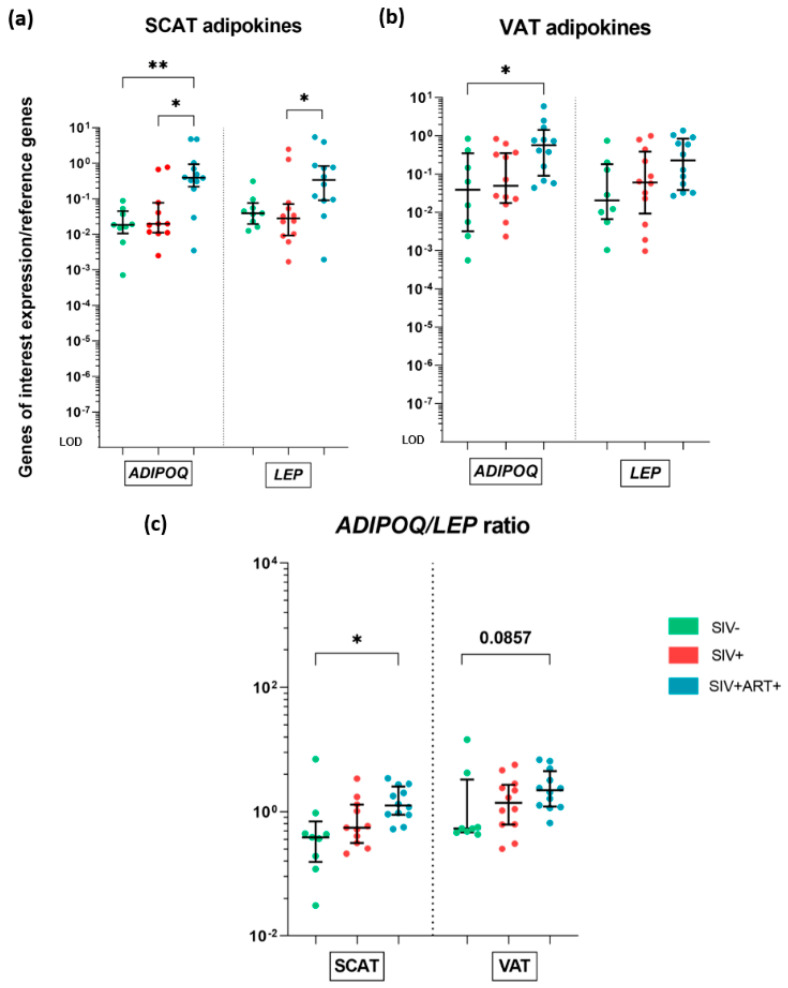
Whole-tissue gene expression analysis of adipokine production by the AT. RT-qPCR analysis of the mRNA expression of genes encoding adiponectin (*ADIPOQ*) and leptin (*LEP*) within the SCAT and VAT, in the three groups of animals: SIV− (green), SIV+ (red), and SIV+ART+ (blue). Expression of the genes of interest was compared with that of the *18S* and *PPIA* housekeeping genes. The LOD was set to the difference vs. the mean level of housekeeping gene expression after 20 cycles, i.e., a 10^− −^^8-^fold difference. (**a**,**b**) mRNA expression of *ADIPOQ* and *LEP* in SCAT (**a**) and VAT (**b**) in the three groups. (**c**) The *ADIPOQ/LEP* ratio in the three groups and within the SCAT (left) and VAT (right). The results are quoted as the median [IQR]. Kruskal–Wallis tests associated with Dunn’s multiple comparison test or Mann–Whitney tests were used for unpaired and paired comparisons, respectively. Only significant results and trends are shown: * *p* < 0.05, ** *p* < 0.005.

**Figure 4 cells-11-03104-f004:**
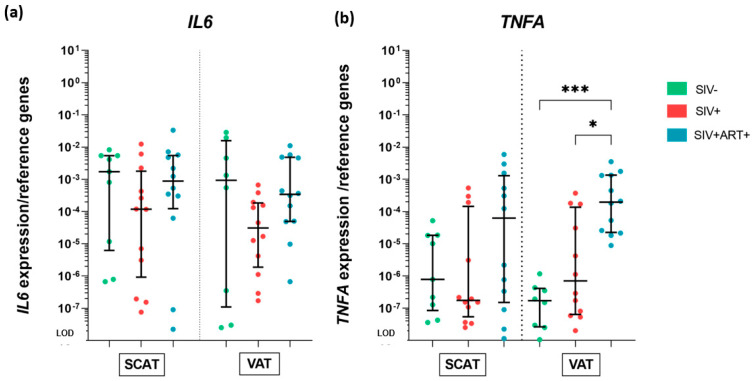
Gene expression analysis of cytokine production by the AT, at the whole-tissue level. RT-qPCR analysis of the mRNA expression of genes encoding the proinflammatory cytokines IL-6 and TNF-α within SCAT and VAT in the three groups of animals: SIV− (green), SIV+ (red), and SIV+ART+ (blue). The expression of the genes of interest was compared with that of the *18S* and *PPIA* housekeeping genes. The LOD was set to the difference vs. the mean level of housekeeping gene expression after 20 cycles, i.e., a 10 ^−^^8^-fold difference. (**a**) Expression of *IL6* in SCAT (left) and VAT (right) in the three groups. (**b**) mRNA expression of *TNFA* in SCAT (left) and VAT (right) in the three groups. The results are quoted as the median [IQR]. Kruskal–Wallis tests associated with Dunn’s multiple comparison test or Mann–Whitney tests were used for unpaired and paired comparisons, respectively. Only significant results and trends are shown: * *p* < 0.05, *** *p* < 0.001.

**Figure 5 cells-11-03104-f005:**
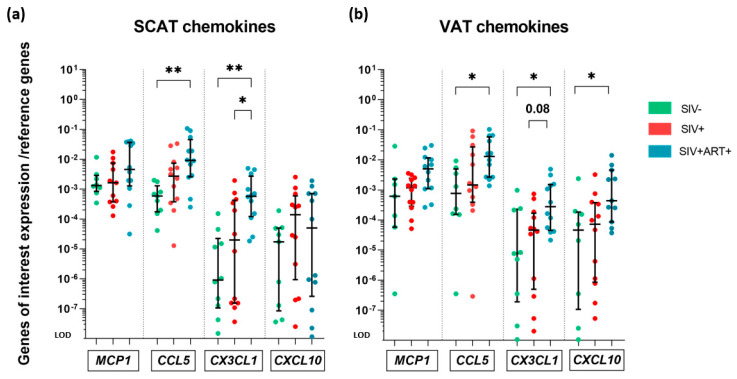
Gene expression analysis of AT chemokine production, at the whole-tissue level. RT-qPCR analysis of the mRNA expression of genes encoding MCP-1, CCL5, CX3CL1 and CXCL10 chemokines within the SCAT and VAT in the three groups of animals: SIV− (green), SIV+ (red), and SIV+ART+ (blue). Expression of the genes of interest was compared with that of the *18S* and *PPIA* housekeeping genes. The limit of detection (LOD) was set to the difference vs. the mean level of housekeeping gene expression after 20 cycles, i.e., a 10^−8^-fold difference. (**a,b**) Expression of *MCP1*, *CCL5*, *CX3CL1* and *CXCL10* within SCAT (**a**) and VAT (**b**) in the three groups. The results are quoted as the median [IQR]. Kruskal–Wallis tests associated with Dunn’s multiple comparison test and Mann–Whitney tests were used for unpaired and paired comparisons, respectively. Only significant results and trends are shown: * *p* < 0.05, ** *p* < 0.005.

**Figure 6 cells-11-03104-f006:**
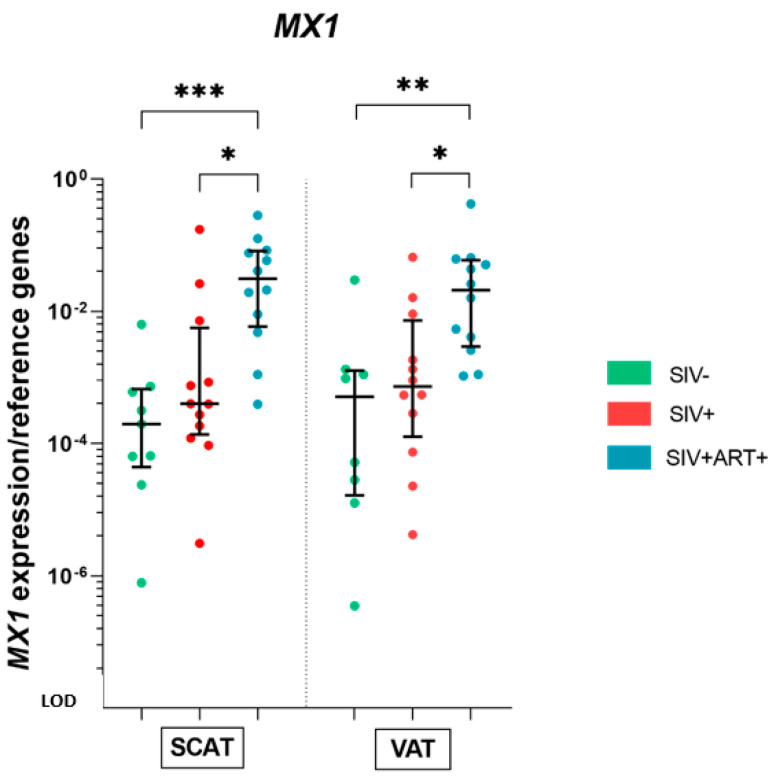
Whole-tissue analysis of type I interferon-stimulated gene expression in AT. RT-qPCR analysis of the mRNA expression of a gene encoding a type I interferon-stimulated gene (*MX1*) within the SCAT and VAT in the three groups of animals: SIV− (green), SIV+ (red), and SIV+ART+ (blue). The expression of the gene of interest was compared with that of the *18S* and *PPIA* housekeeping genes. The LOD was set to the difference vs. the mean level of housekeeping gene expression after 20 cycles, i.e., a 10^−8^-fold difference. Expression of *MX1* in SCAT (left) and VAT (right) for the three groups. The data are quoted as the median [IQR]. Kruskal–Wallis tests associated with Dunn’s multiple comparison test or Mann–Whitney tests were used for unpaired and paired comparisons, respectively. Only significant results and trends are shown: * *p* < 0.05, ** *p* < 0.005, *** *p* < 0.001.

**Figure 7 cells-11-03104-f007:**
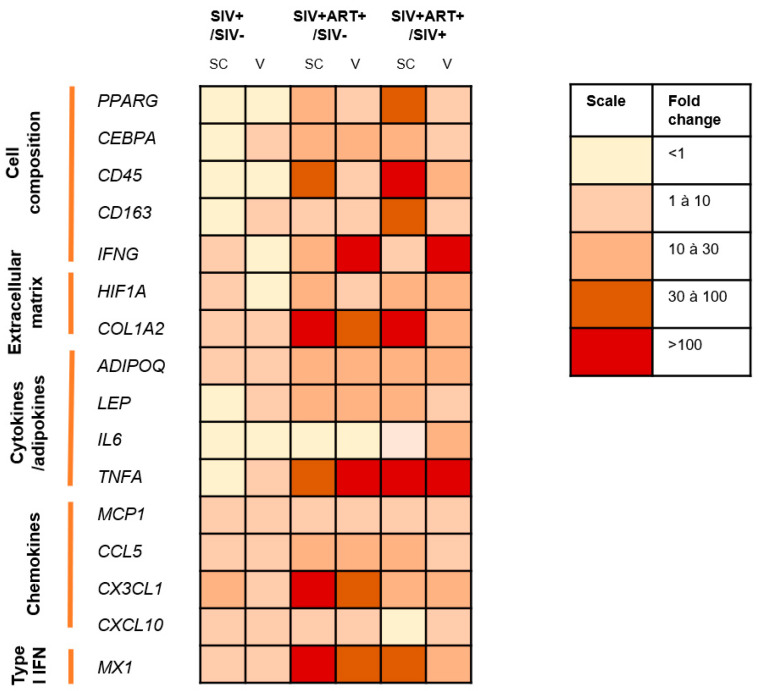
Summary of observed differences in cell composition, inflammatory profile, chemokine expression and interferon-stimulated gene expression within SCAT and VAT. Level of changes (ratio of expression medians) in cell composition, inflammatory profile, chemokine and ISG expression within SCAT and VAT are shown by color gradient for the SIV+/SIV−, SIV+ART+/SIV−, and SIV+ART+/SIV+ groups (expression scale on the right of the table). Each group was compared with the two others.

## Data Availability

The data presented in this study will be openly available, the reference number will be provided after revision and acceptance of the article.

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
