# Peer review of "Prolonged Antiretroviral Treatment Induces Adipose Tissue Remodelling Associated with Mild Inflammation in SIV-Infected Macaques"

_cells, 2022, doi:10.3390/cells11193104_

Round 1

Reviewer 1 Report

In this study the authors examine gene expression patters in adipose tissue of cynomolgus macaques (non-infected, or SIV-infected with or without ART) to examine changes in mRNA expression between these groups.  To assess the effect of SIV infection ± ART on the inflammatory profile of adipose tissue, they focused on the expression of mRNAs involved in metabolic processes (PPARG and CEBPA, as well as markers for fibrosis and hypoxia) and inflammatory pathways (IL-6, TNFα and various cytokines and chemokines).  They demonstrate changes in the expression of the metabolic markers that may indicate abnormalities in the adipose tissue of ART-treated SIV+ macaques (compared to the other groups), and smaller more subtle changes in the inflammatory markers.  This is a thorough study with a lot of comparisons between three groups of animals, examining SCAT and VAT - a lot of work indeed.  While this makes an important contribution in our understanding of the effects of lentiviral infection and ART treatment on adipose tissue, the paper needs to be clearer and more transparent, with improved the data presentation supported by supplementary data tables.

Comments:

-        Given the very subtle changes on inflammatory markers, and lack of change of the IL-6 marker, the title does not seem justified?

Methods:

-        Please provide more transparency of how the analyses were done.  For readers less familiar with qPCR methods, please provide a little more detail how the qPCR reactions were performed, whether multiple genes were assayed together (with two housekeeping genes?), how CT values were converted to relative quantities of mRNA, how exactly the corrections for two housekeeping genes were performed, and whether this was consistent for all the genes of interest.  This would be a more detailed supplementary methods section.

Results:

Section 3.1

-        Please specify at what time point ages and weights of the macaques are reported – at infection? At necropsy?

-        Please specify the measurement of variability (±) – are these standard deviations, standard errors or confidence intervals?

-        Why is the weight in the SIV+ group so very variable (± 3.6 kg - if this is a SD, then 66% of animals are within 3.6 kg of the mean, 7.3 kg, or there is a very significant outlier)?

Section 3.2 – analysing the housekeeping genes.

-      '  We observed a trend towards (30-fold) lower expression of 18S rRNA in the SIV+ART+ group (relative to the SIV- group) in both SCAT and VAT.' (line 201):  What is the comparison here?  When I naively look at the medians on the plot, median expression of 18S rRNA in the SIV- group seems 10-10 and in the SIV+ART+ group 10-13, which looks like a 1000-fold difference, so clearly this trend and p value are calculated in a different way, which is not explained.

Figure 1 – 6 – These are rather hard to read

-        The lines show median and IQR, but are often obscured by the symbols (e.g. Fig 2 Panel (a)).  Ideally the lines should be shown on top of the symbols (the median line must be clearly visible).

-        Please list the number of replicates in the legends – while most SIV+ and SIV+ART+ plot seem to be for 12 data points, the numbers of visible points for SIV- varies between 6 and 10.  Why?

-        While the plots illustrate the scatter of the data points, it would be really helpful to also list the actual values of medians and IQR as supplementary data

-        There could be more consistency in reporting comparisons, e.g. sometimes P-values are listed (with a *, **, *** system, or as numbers in situations where P was considered non-significant. Supplementary data tables showing all medians and IQR values, comparisons (ratios) of the medians and exact P-values would be really helpful.

-        In Figure 4(b) SCAT, order of samples or colour code (SIV+ vs SIV+ART+) are reversed?  Please correct

A lot of the comparisons are based only on the P-values, using a strict binary interpretation of significance as P <0.05.  Although common in lab science, coding of P-values as *, ** or *** and only reporting non-significant P-values is misleading, especially as P-values based on non-parametric methods (ranks) are not generally very robust.  P values represent the probability that a result as extreme as the observed may occur under the null hypothesis.  Thus P = 0.05 corresponds to a 1/20 chance, P = 0.0683 (Fig 2(b)) a 1/15 chance, while a * could, theoretically, refer to P = 0.045, a 1/22 chance.  The multiple pairwise comparisons may require further stringency rules, as some * results might be observed purely by chance?

-        Which markers are considered representative of fibroblasts (line 244)?

Looking at specific data:

-        Although the description of the results generally agrees with the plots, but I find it hard to see/don’t always agree with the ratios of expression levels quoted, especially when x-fold differences are described.  For example

o   Line 269 - >2 log-fold difference in medians seems to apply to SCAT but not VAT?

o   Figure 2(c) for PPARG/COL1A2 ratios – calculating ratios does increase the spread, and all ‘significant’ P values are only presented as * - overall it seems there is little evidence of large differences between the groups of animals.

o   Figure 2(d) for HIF shows a similar trend, but conclusions about different effects in SCAT and VAT are probably not justified (P values shown as * are difficult to interpret)

o   Overall, the spread of data points (highly heterogenous gene expression) needs generally more careful conclusions

Fig 7:  What does the colour indicate?  While the spread of data probably does not allow calculation of ratios in mRNA expression (as ‘significantly (16-fold) greater’ -line 372 or ‘much higher levels (by a factor of up to 633)’ line 374) the authors might consider showing a heat map coloured approximately for log differences of higher/lower expression to be more informative.  Please remove the spell-check marks in the marker classification on the left.

-        The Discussion section is well-written and thoughtful, and highlights limitations.  In the conclusions, specify that the points raised in lines 619-625 – very good summaries! – are based on this together with other published studies.

Reviewer 2 Report

to put this paper into clinical perspective, there is concerns in the clinical community that ARt treatment results in weight gain, some combinations more than others.  it would be good to address these issues in the paper, and mention which ART combinations were used in this study

Author Response

We thank the reviewer for his/her positive comments and we modified the manuscript as suggested.

to put this paper into clinical perspective, there is concerns in the clinical community that ARt treatment results in weight gain, some combinations more than others.  it would be good to address these issues in the paper.

It is indeed a clinical concern and we modified the manuscript to highlight this aspect. To note, we did not observe any weight gain in the treated animals and thus we did not further discuss this point.

and mention which ART combinations were used in this study:

We developed the description of the ARV treatments by introducing this sentence: “This combination is currently used in clinical context and includes 2 nucleoside reverse transcriptase inhibitor and 1 integrase strand transfer inhibitor (INSTI)” Line 125.

Reviewer 3 Report

the paper describe the adipose inflammation in SIV infection.

The paper  is well done and clear enough.

it would be useful for the authors to explain the reason for choosing TDF / FTC / DTG as the therapeutic standard for SIV.

it would be useful for the authors to highlight the role of antiretroviral therapy in the inflammatory state

a better explanation of the connection between inflammation and adipocyte activity with and without antiretroviral therapy would be useful in order to understand the role of siv and therapy

Author Response

We thank the reviewer for his/her positive comments and we modified the manuscript as suggested.

  • It would be useful for the authors to explain the reason for choosing TDF / FTC / DTG as the therapeutic standard for SIV: We have now introduced the following sentence ““This combination is currently used in clinical context and includes 2 nucleoside reverse transcriptase inhibitor and 1 integrase strand transfer inhibitor (INSTI)” Line 125.
  • it would be useful for the authors to highlight the role of antiretroviral therapy in the inflammatory state. As mentioned by reviewer 1 the level of inflammation is relatively mild and thus we did not want to further emphasize the role of ARVs in the inflammatory state. However, we have changed the title to highlight the role of ARVs on adipose tissue remodeling
  • a better explanation of the connection between inflammation and adipocyte activity with and without antiretroviral therapy would be useful in order to understand the role of siv and therapy: We agree with the reviewer, our results do suggest that there is a different link between adipocyte activity/dysfunction and inflammatory state in the context of obesity and in the context of ART-treated infection. In the context of the SIV+ART+ we observed clear AT remodeling associated with mild inflammation. It is tempting to speculate that the 2 processes involve different pathways of adipose tissue alterations; it could also be due to the duration of the two phenomena (several years for obesity compared to 2 years of treatment in our study). This is an aspect that we would like to further discuss but we do not have enough data to support it. We have introduced a short sentence on the subject Line 571.